# Cumulative effects of weakly repressive regulatory regions in the 3' UTR maintain PD-1 expression homeostasis in mammals

Xiaoqian Lai [1,3], Rong Li [1,3], Panpan Wang [1], Meng Li [1], Chenxi Xiao[2], Qiang Cao [1], Xin Li [1,4✉] & Wenxue Zhao [1,4✉]

PD-1 has become a common target for cancer treatment. However, the molecular regulation of PD-1 expression homeostasis remains unclear. Here we report the PD-1 3' UTR can dramatically repress gene expression via promoting mRNA decay. Deletion of the PD-1 3' UTR inhibits T cell activity and promotes T-ALL cell proliferation. Interestingly, the robust repression is attributable to cumulative effects of many weak regulatory regions, which we show together are better able to maintain PD-1 expression homeostasis. We further identify several RNA binding proteins (RBPs) that modulate PD-1 expression via the 3' UTR, including IGF2BP2, RBM38, SRSF7, and SRSF4. Moreover, despite rapid evolution, PD-1 3' UTRs are functionally conserved and strongly repress gene expression through many common RBP binding sites. These findings reveal a previously unrecognized mechanism of maintaining PD-1 expression homeostasis and might represent a general model for how small regulatory effects play big roles in regulation of gene expression and biology.

[1] Molecular Cancer Research Center, School of Medicine, Shenzhen Campus of Sun Yat-sen University, Sun Yat-sen University, Shenzhen 518107, China. [2] Undergraduate Program in Medicine, School of Medicine, Shenzhen Campus of Sun Yat-sen University, Shenzhen 518107, China. [3] These authors contributed equally: Xiaoqian Lai, Rong Li. [4] These authors jointly supervised this work: Xin Li, Wenxue Zhao. ✉email: lixin253@mail.sysu.edu.cn; zhaowx5@mail.sysu.edu.cn

The human programmed cell death-1 (PD-1), encoded by the *PDCD1* gene, is an inhibitory receptor mediating central and peripheral immune tolerance and immune exhaustion[1]. On resting naive T cells, as well as in certain populations of developing thymocytes, PD-1 is expressed at low basal levels to maintain immune tolerance[2,3]. Mutation or deficiency in PD-1 has been shown to be associated with disease progression in multiple human autoimmune disorders[4]. During chronic viral infections, high levels of PD-1 is expressed on the surface of T cells, NK cells, B cells, macrophages and subsets of DCs[5]. Constitutive expression of PD-1 upon ligation to its ligand PD-L1 leads to functional exhaustion. Exhausted CD8[+] T cells are unable to secrete normal amounts of cytokines, proliferate, or perform immune functions such as initiating cellular cytotoxicity, thus remarkably restraining the tumor-specific immune response[1,6]. Therefore, the maintenance of PD-1 expression homeostasis is essential in maintaining normal immune functions.

PD-1 expression is tightly and dynamically regulated to adapt to either transient or chronic antigenic stimulations. Earlier studies focusing on the transcriptional programs revealed two conserved DNA regions (CR-B and CR-C) associated with PD-1 activation[7]. These elements contain multiple binding sites for activating transcription factors (TFs), including AP-1[8], NFAT[7], FoxO1[9], NF-κB[10], and Notch[11]; other TFs, such as T-bet and Blimp-1 are demonstrated to be transcriptional inhibitors for PD-1[12,13]. Epigenetic modifications are also widely involved in the control of gene transcription and influence immune cell fate. 5-hydroxymethylcytosine (5hmC) in the CR-B and CR-C regions was negatively correlated with PD-1 expression[14]. Histone modifications such as H3K9ac and H3K27ac acetylation can affect PD-1 transcriptional activity by changing chromatin accessibility[15]. Several microRNAs including miR-4717, miR-374b, miR-28, and miR-138 have been identified to repress PD-1 expression by directly binding to its 3' UTR[16–19]. Interestingly, miRNA-mediated PD-1 repression can be relieved by cicrRNAs serving as miRNA sponges[20]. In addition, there is increasing evidence demonstrating the importance of posttranslational modification that controls the PD-1 expression. For instance, FBXO38 can function as an E3 ligase and promote proteasome-mediated PD-1 degradation through Lys48-linked poly-ubiquitination[21]. TOX could stabilize PD-1 by binding to PD-1 to prevent lysosomal-mediated degradation[22]. A more recent work showed that KLHL22 acts as an adaptor of the Cul3-based E3 ligase to promote the degradation of PD-1 before its transport to the cell surface[6].

Despite these great achievements, it remains unknown how other genetic information contributes to the intricate regulation that maintains PD-1 expression homeostasis. The 3' UTR is a type of noncoding sequence that is localized at the 3' end of messenger RNAs (mRNAs). Tremendous work has been done to demonstrate the critical roles of 3' UTRs in the regulation of many aspects of biological processes through types of *cis*-regulatory elements contained in 3' UTRs. These elements can be either linear or structural units varying from several to decades of nucleotides or longer in size, and serve as binding sites for numerous RNA binding proteins (RBPs) and noncoding RNAs[23]. The binding of such *trans*-acting factors can modulate biological complexity by influencing mRNA decay, subcellular localization, translation rates, and other aspects of mRNA biology[24–27]. A well-known type of *cis*-regulatory element is the AU-rich element (ARE) which was first found in the *c-fos* 3' UTR[28]. A typical ARE consists of one or more of UAUUUAU repeats[29], which, recognized by different RBPs, leads to diverse consequences in an RBP-dependent manner. For example, TTP binding to the ARE in the PD-L1 3' UTR destabilizes mRNA; inhibition of TTP by MEK signaling promotes lung and colorectal tumors owing to the increase in PD-L1 mRNA stability[24]. In contrast, HuR binding could stabilize many ARE-containing mRNAs[30]. Other characterized elements in 3' UTRs include GU-rich elements[31], CU-rich elements[32], and higher structural elements such as hairpins and stem-loops[33].

Since the recent finding that genes with rapidly evolving 3' UTRs are markedly associated with metabolism and immune responses[23], we first compared the conservation of 3' UTRs of 21,050 genes of 99 vertebrates, with the finding that the PD-1 3' UTR evolved much rapidly compared to most 3' UTRs of other genes. Reporter assays demonstrated that the PD-1 3' UTR dramatically reduced protein production and mRNA abundance, which was tightly coupled with mRNA decay in multiple cell lines, including T cells. These results were further recapitulated in native cellular and genomic contexts by disruption of the endogenous PD-1 3' UTR using CRISPR-Cas9. By comprehensively mapping through fragmentation, truncation, and mutation, we found that the regulatory activity of the PD-1 3' UTR was controlled by many regions with weak regulatory activity instead of one or few dominant elements as we initially speculated. To explore the cooperating *trans*-acting factors, we identified a set of RBP binding sites within the PD-1 3' UTR. Several RBPs including MBNL1, IGF2BP2, RBM38, SRSF7, and SRSF4 were validated to control the PD-1 expression through the 3' UTR. Functionally, loss of MBNL1 or the whole 3' UTR could significantly promote cell proliferation or inhibited T cell activation. Further, we measured the regulatory effect of the 3' UTRs of several primates and the mouse, with the finding that all these 3' UTRs surprisingly exhibited similar inhibitory effect, which was coupled with many conserved RBP binding sites. In addition, we demonstrated that the 3' UTRs could evolve through the gain or loss of specific RBP binding sites to adapt to cellular contexts.

## Results

**The PD-1 3' UTR repressed reporter gene expression by promoting mRNA decay.** The human PD-1 3' UTR is made up of 1,174 nucleotides, longer than its coding region (867 nts). To explore the revolutionary relationship of PD-1 3' UTRs of species, we systematically analyzed the sequence conservation of 21,050 protein-coding genes from 99 vertebrates. While the conservation score of PD-1 CDS was ranked at a medium position, 55% (11,621/21,050), the PD-1 3' UTR was ranked at 93% (19,609/21,050) (Fig. 1a, Supplementary Data 1), suggesting that the PD-1 3' UTR evolves much rapidly than most 3' UTRs of other genes. Given that rapid evolution of sequences could be due to either relaxation of functional constraint or positive selection, we decided to test whether the PD-1 3' UTR is functional in the regulation of gene expression. We cloned it into the reporter BTV (Fig. 1b) that was previously established to quantify the regulatory effect of 3' UTRs[29]. The flow cytometric analysis showed that the PD-1 3' UTR dramatically reduced reporter protein production compared with the control in WiDr cells (a colon cancer cell line) (Fig. 1c). Since the regulatory activity of 3' UTRs relies on the *trans*-acting factors such as RBPs and noncoding RNAs which vary across cell types, we tested the PD-1 3' UTR in three other human cell lines: Jurkat (T lymphoblast), BEAS-2B (airway epithelium), and HelaS3 (cervical carcinoma) cells. Surprisingly, the PD-1 3' UTR resulted in remarkable decrease in reporter protein in all cell lines tested, to 19.4% in BEAS-2B, 10.8% in HelaS3, 13.5% in Jurkat, and 13.9% in WiDr compared with the control (100%) (Fig. 1d), implying that similar *trans*-acting RBPs might interact with the 3' UTR in these cell lines. To test whether the decrease in protein level was the result of the decrease in mRNA level, we used qRT-PCR to measure the steady-state EGFP

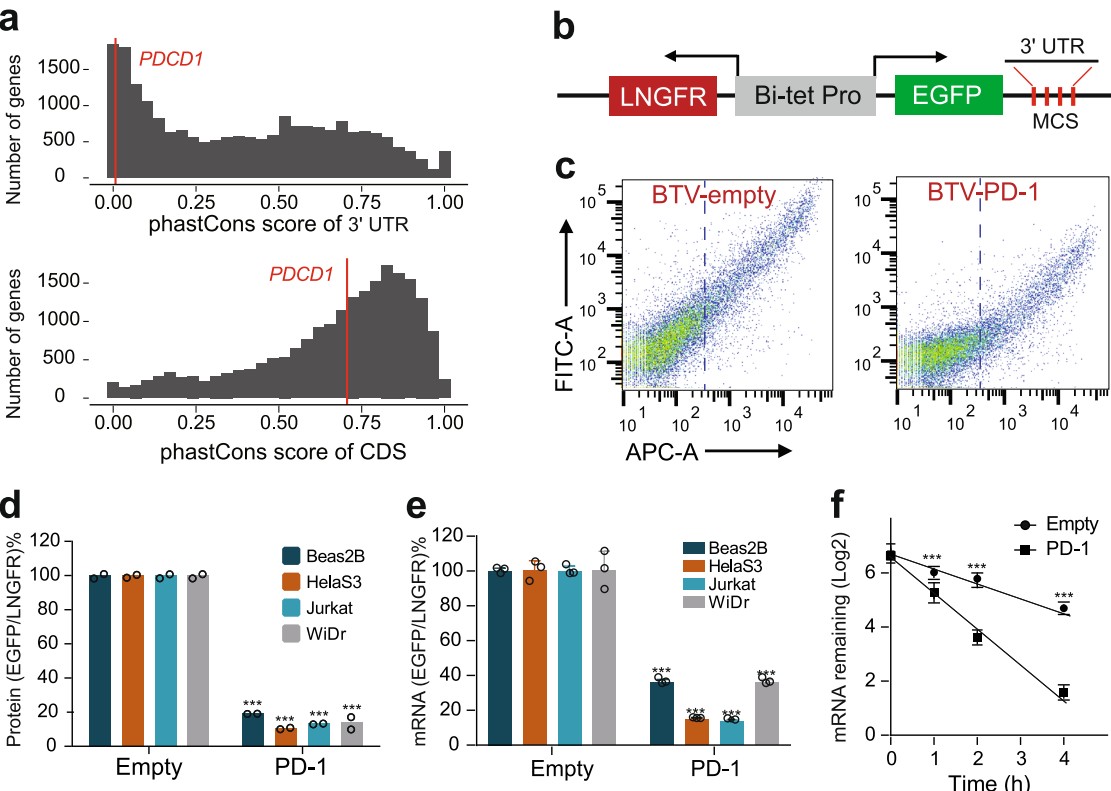

**Fig. 1 The conservation and regulatory effects of the human PD-1 3' UTR. a** Conservation scores of the 3' UTRs (the upper panel) and the coding regions (CDS, the lower panel) of 21,050 genes of 99 vertebrates. The red vertical line denotes the human *PDCD1* gene. Greater phastCons scores indicate higher conservation levels. **b** The schematic diagram of the BTV reporter. The reference gene *LNGFR* and the reporter gene *EGFP* are driven by the bidirectional tetracycline-regulated promoter. The expression of the *EGFP* but not the *LNGFR* can be affected by the 3' UTR cloned downstream at the multiple cloning sites (MCS). The ratio of EGFP/LNGFR reflected the activity of the cloned 3' UTR. **c** Flow cytometric analysis of the effect of the PD-1 3' UTR on reporter gene expression in WiDr cells. The LNGFR expressed on the cell surface was stained with the APC-conjugated antibody; the EGFP was detected via FITC channel. Antibody-stained but untransduced cells were used to gate the LNGFR-negative cells. The cells right to the vertical dotted line represented LNGFR positive cells which were used to calculate the ration of EGFP/LNGFR. **d**, **e** The effect of the PD-1 3' UTR on reporter protein (**d**, n = 2) and reporter mRNA (**e**, *n* = 3) in four cell lines. **f** The effect of the PD-1 3' UTR on reporter mRNA decay in Jurkat cells; *n* = 3. All values represent mean ± s.d. ***, *p* < 0.001 by Student's t-test.

mRNA level (Fig. 1e). As expected, the PD-1 3' UTR significantly reduced mRNA abundance in all four cell lines. Further, the time-course-based mRNA decay assay in Jurkat cells demonstrated that the PD-1 3' UTR promoted more rapid reporter mRNA decay, with the half-life of 2.17 h for the empty BTV control and 0.79 h for the PD-1 3' UTR (Fig. 1f). We therefore concluded that the PD-1 3' UTR could constitutively repress gene expression by promoting mRNA degradation.

**Consistent repressive effect of the PD-1 3' UTR in native genomic context.** The BTV reporter is an artificial DNA construct lacking native genetic information, such as the promoter, the 5' UTR, and the introns that might affect the regulatory effect of the PD-1 3' UTR. To examine this possibility, we expressed two CRISPR gRNAs in MOLT-4 cells, a T lymphoblast cell line derived from an acute lymphoblastic leukemia (T-ALL) patient, which constitutively expresses PD-1. The two gRNAs respectively cleaved after the stop codon and before the polyA signal to delete the DNA sequence encoding the major portion of the PD-1 3' UTR, thus resulting in a native PD-1 transcripts except for the lack of the 3' UTR[34] (Fig. 2a and b). Flow cytometric analysis showed that removal of the 3' UTR substantially increased the endogenous PD-1 protein expressed on the surface of the cells as expected from the reporter assay (Fig. 2c). Similar to the effect on protein, lack of the PD-1 3' UTR also increased endogenous

mRNA level markedly (Fig. 2d). We further measured the mRNA decay rate after stopping endogenous transcription in MOLT-4 cells with actinomycin D, with the result that deletion of the 3' UTR slowed down PD-1 mRNA decay by increasing mRNA half-life from 1.1 h to 1.8 h (Fig. 2e). Since the basic function of PD-1 is inhibition of T lymphocyte activation, we wondered whether PD-1 elevation due to removal of the 3' UTR could potentially inhibit T cell activity. Quantification of IL-2 and CD4 (two of key markers for T cell activation) mRNA revealed that deletion of the PD-1 3' UTR markedly reduced expression of both cytokines in 3' UTR-deleted MOLT-4 cells compared with control cells (Fig. 2f and g), suggesting the role for PD-1 3' UTR in the maintenance of T cell activation. The second biological effect of PD-1 is promoting cell proliferation in some types of cancer cells. The significant increase in Ki67 (a proliferation marker) expression in 3' UTR-deleted MOLT-4 cells compared to the control indicated the ability of PD-1 3' UTR to inhibit T-ALL cell proliferation (Fig. 2h), which was further supported by time-course based cell proliferation assays (Fig. 2i). We were next interested in the mechanism by which deletion of PD-1 3' UTR increased the proliferation rate of MOLT-4 cells. MOLT-4 cells are cancer cells derived from a patient with T-cell acute lymphoblastic leukemia. Recent studies showed that intrinsic PD-1 expression in many types of cancer cells can promote cell proliferation, including melanoma[35], pancreatic cancer, hepatocellular carcinoma[36], lung cancer[37], and skin cancer[38], and one of the mechanisms is

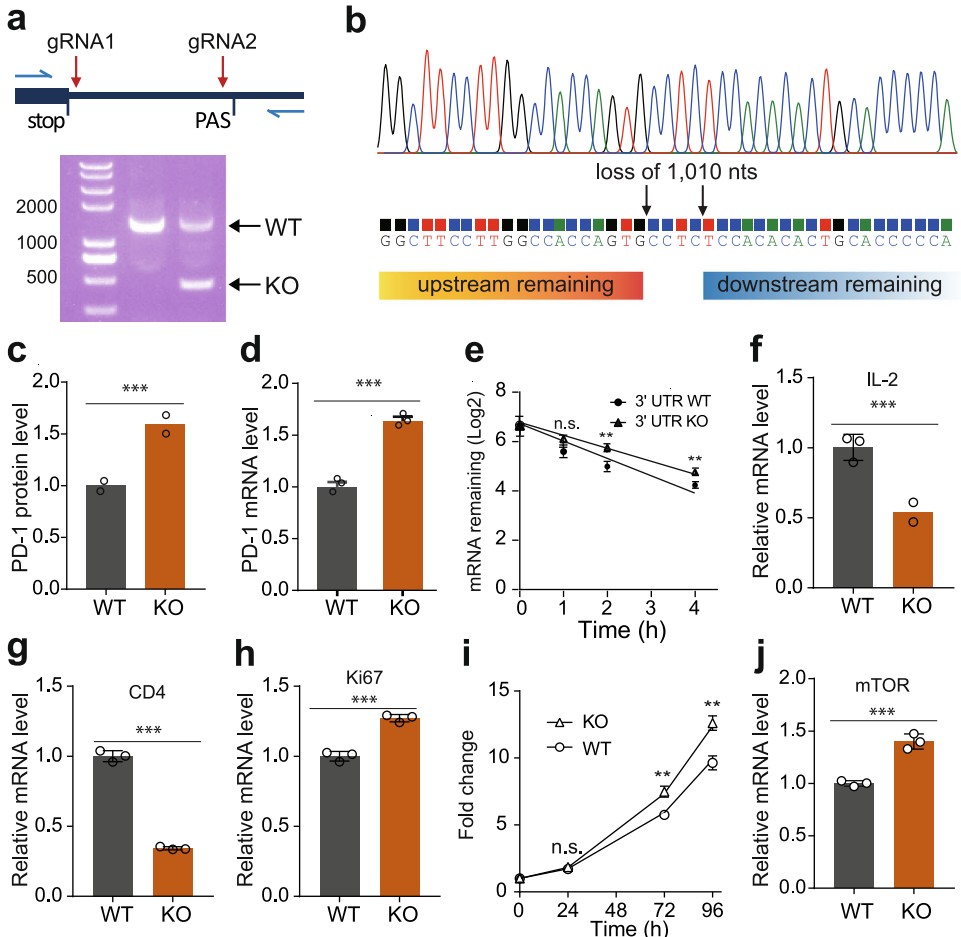

**Fig. 2 The effect of the PD-1 3' UTR on endogenous PD-1 expression, T cell activation, and cell proliferation. a** Depletion of the genomic DNA sequence encoding the PD-1 3' UTR using two CRISPR gRNAs. gRNA$_{1/2}$, the position of the two gRNAs; stop, stop codon; PAS, polyA signal; blue arrows denote the position of PCR primers. The gel picture was PCR detection of the PD-1 3' UTR before (the 2$^{nd}$ lane) and after (the 3$^{rd}$ lane) CRISPR-Cas9 deletion. **b** Confirmation of the deletion of the PD-1 3' UTR by Sanger sequencing. **c-e** The effect of depletion of the PD-1 3' UTR on the level of PD-1 protein (**c**), steady state PD-1 mRNA (**d**), and PD-1 mRNA decay endogenously (**e**); $n = 3$. **f-h** The effect of depletion of the PD-1 3' UTR on the mRNA level of IL2 (**f**), CD4 (**g**), Ki67 (**h**); $n = 3$. (**i**) CCK8 assay for quantification of MOLT-4 cell proliferation; $n = 3$. **j** The effect of depletion of the PD-1 3' UTR on the mRNA level of mTOR; $n = 3$. KO: Knockout of the PD-1 3' UTR, WT: 3' UTR-intact cells. All values represent mean ± s.d. ***, $p < 0.001$, **, $p < 0.005$ by Student's t-test, n.s., no significant difference.

up-regulation of mTOR which modulates some of genes related to cell proliferation[35]. We therefore detected mTOR expression in PD-1 3' UTR-deleted MOLT-4 cells. Indeed, the mTOR expression was increased by 40%, suggesting PD-1 3' UTR deletion promoted MOLT-4 cell proliferation through the oncogenic pathway. However, the detailed mechanism could be more complicated and further investigation would be valuable. To be emphasized, we failed to obtain homozygous 3' UTR knockout cells, probably because MOLT-4 cells are of tetraploid karyotype so that it was difficult to completely delete all copies of 3' UTR-DNAs with CRISPR-Cas9 (Fig. 2a). Considering this, our results might underestimate the effect of the 3' UTR on endogenous PD-1 expression (Fig. 2c-e). Collectively, our results demonstrated that the PD-1 3' UTR could repress gene expression by promoting mRNA decay in both reporter and native contexts. Functionally, the PD-1 3' UTR could participate in the maintenance of T cell activation and inhibition of ALL cell proliferation.

**Systematic mapping of active regulatory regions in the PD-1 3' UTR.** Next, we sought to map the key regions that were

responsible for the inhibitory activity of the PD-1 3' UTR. Considering that its activity was such strong (Fig. 1d), we initially speculated there existed one or few dominant elements with strong inhibitory activity. To test the hypothesis, we divided the PD-1 3' UTR into three fragments with an overlap between any two adjacent ones, P1 (1-398), P2 (349-795), P3 (711-1174), and assessed their regulatory activity in BEAS-2B and WiDr cells after cloning in the BTV reporter (Fig. 3a). Flow cytometric analysis showed that all three fragments had pronounced inhibitory activity (Fig. 3b). Interestingly, while P1 and P3 showed fair inhibitory activity (30-50%), P2 dramatically reduced the reporter gene expression, less than 10% (Fig. 3b), suggesting P2 represented the major regulatory activity of the PD-1 3' UTR. We therefore further divided P2 into four overlapping fragments: P2a, P2b, P2c, and P2d (Fig. 3a). Unexpectedly, each of the four fragments exhibited only modest inhibitory activity in comparison to P2 (Fig. 3c), suggesting that the inhibitory activity of the PD-1 3' UTR was not governed by one or few dominant elements, but rather by multiple regions with weak activity.

In order to systematically localize active elements, we tested a collection of truncated sequences derived from the P2 fragment with a spacer of ~60 nts from either 5' or 3' ends (Fig. 3d). As

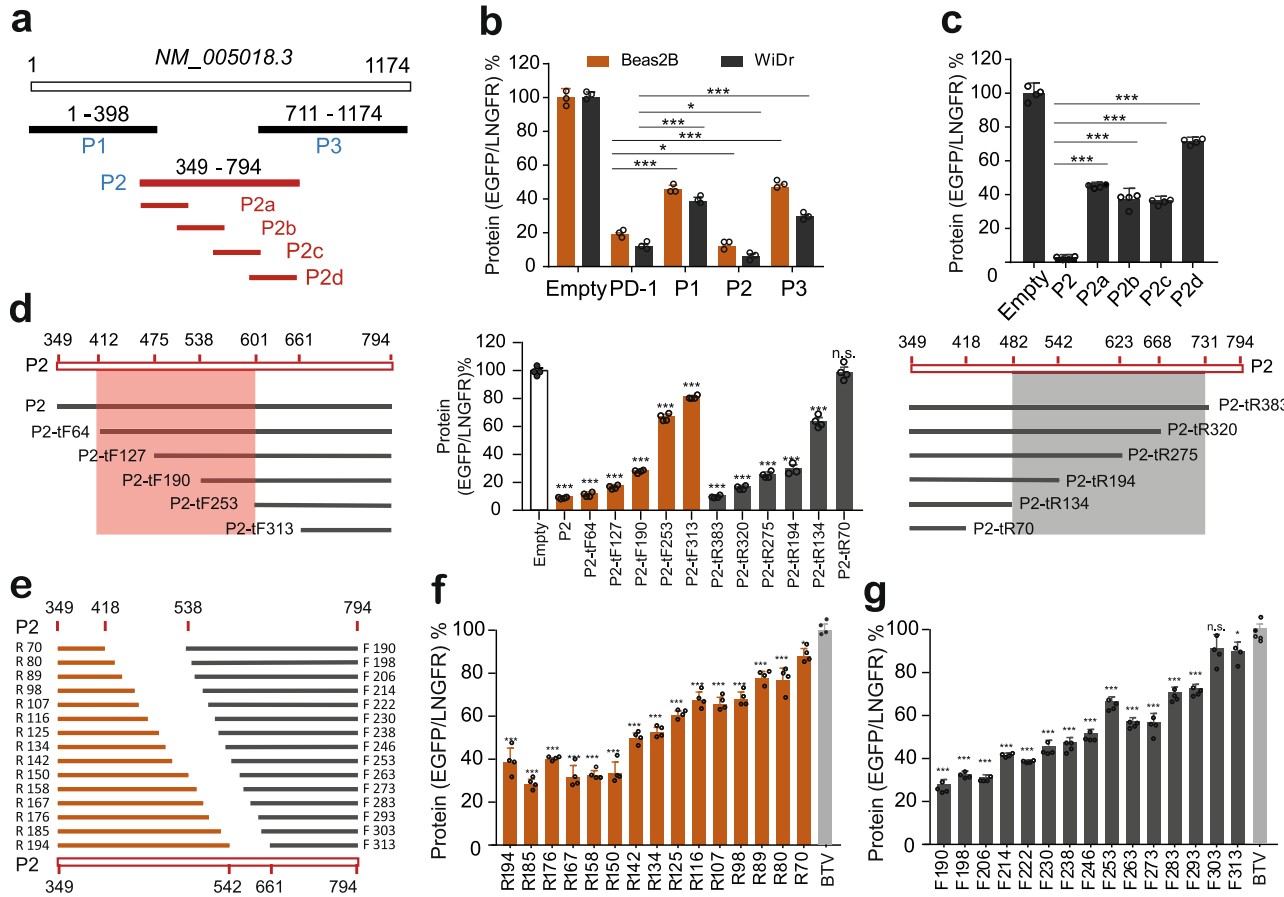

**Fig. 3 Mapping of active regulatory regions in the PD-1 3' UTR by fragmentation and truncation. a** Design of overlapping fragments of the PD-1 3' UTR. **b** The effect of P1, P2, and P3 on reporter protein in BEAS-2B and WiDr cells; $n = 3$. **c** The effect of P2a, P2b, P2c, and P2d on reporter protein in WiDr cells; $n = 4$. **d** Measurement of the activity of the truncations with 60 nts forwardly and reversely; $n = 4$. **e** Design of 10-nt truncations for the most active regions. **f**, **g** Quantification of the effect of 10-nt truncations on reporter protein; $n = 4$. All values represent mean ± s.d. *, $p < 0.05$, ***, $p < 0.001$ by one-way ANOVA and Dunnett's test.

seen in Fig. 3d, the removal of any regions led to reduced inhibitory activity, and the longer the truncation was, the more activity was lost, suggesting multiple elements spread across the P2. In addition, the results showed that two regions (412–601 and 482–731) appeared to be the most active (Fig. 3d). We therefore further narrowed the resolution for these two regions by truncation spaced by 8 ~ 10 nts (Fig. 3e). By examining the set of resulting fragments in the reporter, we did not see any of single 8 ~ 10-nt regions that dramatically affected the activity. Instead, the inhibitory activity was lost bit by bit with truncation lengthening (Fig. 3f and g), suggesting the regulatory activity of the PD-1 3' UTR was the result of the cumulation of many weak regulatory regions.

**Cumulated weak regulation could facilitate the maintenance of gene expression homeostasis against mutagenesis.** To fine map the functional elements that were responsible for the regulatory activity of the P2 fragment, we further performed the mutational analysis in three key regions spanning 393–458, 499–578, and 629–728, respectively (Fig. 4a). In specific, we designed consecutive 6-nt (only for the region 413-418) or 10-nt mutations by A-U and C-G replacement in these regions (Fig. 4a). Reporter assays revealed that many of mutations led to slight or moderate increase in protein production compared to the wild type sequence (Fig. 4b), indicating potential weak regulatory elements or RBP binding sites. To investigate whether individual regulatory regions affecting gene expression was consistent with the full length did, we

measured the effect of the region 413-418 on reporter protein, steady-state mRNA, and mRNA decay rate. As expected, this region reduced mRNA abundance and protein production that were associated with more rapid mRNA decay (Fig. 4c-e). These results further supported the notion that the strong inhibitory activity of the PD-1 3' UTR was derived from many weak regulatory elements or RBP binding sites. This inspired us to think why the PD-1 3' UTR evolved to contain so many weak regulatory elements instead of one or few dominant elements to repress PD-1 expression. We proposed that co-repression by many weak elements could better buffer deleterious mutagenesis than a single strong element, because a single nucleotide mutation in a strong element (such as an ARE) may dramatically increase gene expression, which would be a disaster in the case of PD-1 because elevated PD-1 level is one of the major causes for T cell exhaustion. As shown in Fig. 4f, a single A to C mutation in the *CDKN2D* ARE dramatically increased the reporter protein level to 1.72 folds. In contrast, 10-nt mutations in the PD-1 3' UTR caused only subtle or no effect (Fig. 4b), which is particularly important because a stably low level of PD-1 is essential to maintain normal immune status. We therefore proposed a general model describing how cumulated weak regulation benefits the maintenance of gene expression homeostasis (Fig. 4g). To further support our proposed model, we examine the effect of sixteen natural mutations or variations occurred within the active regions of the PD-1 3' UTR. As expected, 14/16 did not significantly change reporter gene expression; 2/16 showed just slight effect (Fig. 4h).

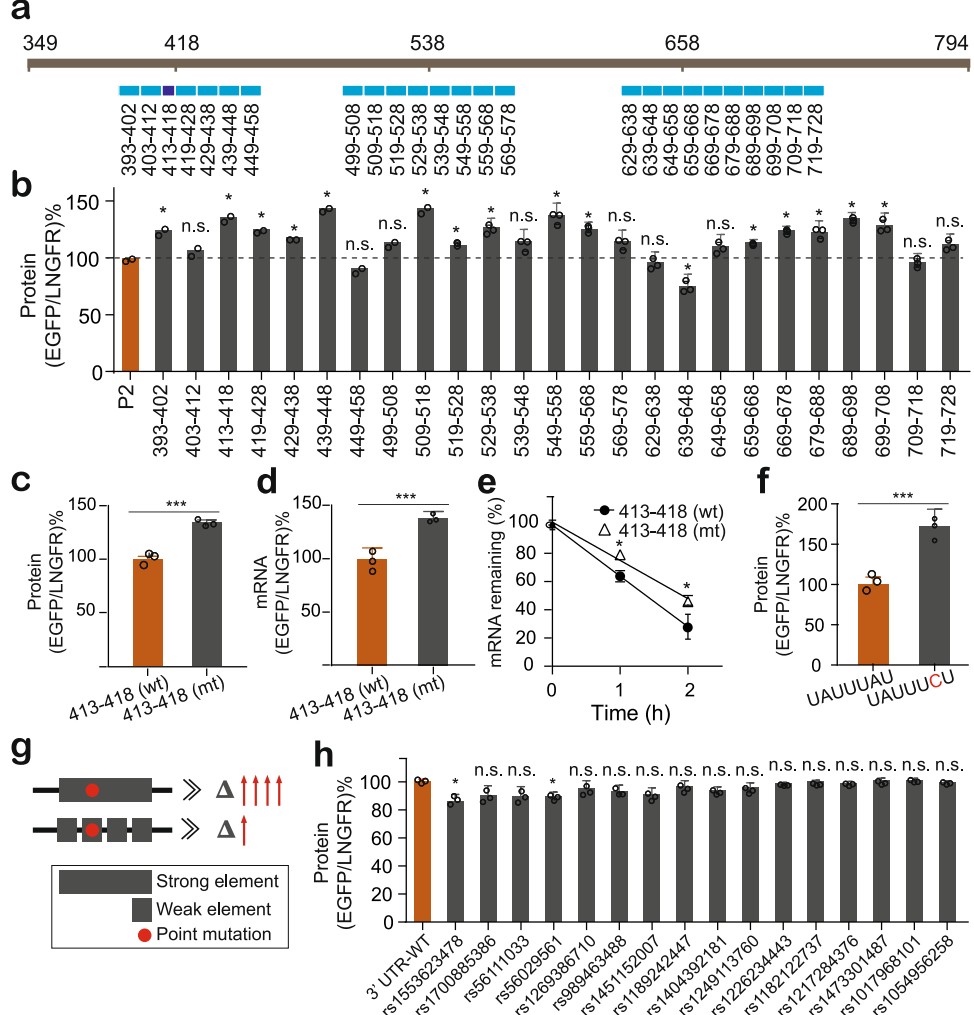

**Fig. 4 The effect of artificial mutations and natural variations on reporter gene expression. a** Design of 10-nt or 6-nt mutations within active regions. **b** Effect of 10- or 6-nt mutations on reporter protein; $n = 2$ or 3. **c–e** Effect of the region 413-418 on reporter protein (**c**), steady state mRNA (**d**), and mRNA decay (**e**); $n = 3$. **f** Effect of a single A to U mutation within the ARE contained in the *CDKN2D* 3' UTR on reporter gene expression; $n = 3$. **g** The proposed model describing how cumulated weak regulation outperforms single strong elements for maintenance of gene expression homeostasis against mutagenesis. **h** Effect of naturally occurred variations on reporter protein expression; $n = 3$. All values represent mean ± s.d. *, $p < 0.05$, ***, $p < 0.001$ by one-way ANOVA and Dunnett's test (**b**, **h**) and Student's t-test (**c–f**); n.s., no significant difference.

**The inhibitory activity of the PD-1 3' UTR depended on multiple RNA-binding proteins.** Since the regulatory activity of 3' UTRs relies on the interaction with *trans*-acting factors such as RBPs and noncoding RNAs. We first examined whether the critical regions overlapped with known or predicted miRNA targeting sites. As shown in Fig. 5a, only one (miR-28) out of twelve miRNA targets overlapped with an active region 499–578, suggesting that miRNAs were not the major regulators responsible for the repressive activity of the PD-1 3' UTR. We therefore sought to reveal which RBPs were involved in the regulation mediated by the PD-1 3' UTR. The RBPmap was used to predict RBP binding sites[39], which yielded a list of 86 RBPs with a total of 803 predicted binding sites in the PD-1 3' UTR. Although not able to validate all these predicted RBPs, we selected 12 for further investigation (Fig. 5b). These proteins were selected based on their previously known function particularly in control of mRNA stability. In specific, DAZAP1 is a component of complexes that are crucial for the degradation and silencing of mRNA[40]; FMR1 binds 3' UTRs to contributes to maternal RNA degradation[41]; FXR1 and FXR2 are involved in the transport, translation, and degradation of mRNA[42,43]; YBX1 can decrease mRNA stability of

*Pink1* and *Prkn*[44]; IGF2BP2/3 can modulate mRNA stability and translation[45]; Rbm38 is required for p63 mRNA degradation[46]; members of the SRSF family such as SRSF3 play roles in splicing and regulate additional aspects of RNA metabolism like alternative polyadenylation, mRNA export[47]. Notably, eight of these RBPs are conserved between human and mice, which are FMR1, FXR2, IGF2BP2, IGF2BP3, RBM38, SRSF2, SRSF5, SRSF9. Additionally, we included 3 nonpredicted RBPs (FUS, RBM42, TARDBP) as controls (Fig. 5b). We next used CRISPR-Cas9 to knock out these RBP genes in MOLT-4 cells and monitored the change in the PD-1 protein level. Flow cytometric analysis demonstrated that 9 out of the 15 RBPs slightly or moderately elevated the PD-1 protein level after knockout (Fig. 5c), suggesting there could be many RBPs that potentially regulate PD-1 expression, although only a small portion of predicted RBPs were tested. To determine whether these nine RBPs affect PD-1 expression through the 3' UTR, we infected the knockout cells with BTV reporter containing the PD-1 3' UTR and measured the change in EGFP expression, with the result that four (IGF2BP2, RBM38, SRSF7, and SRSF5) out of the nine RBPs were found to increase the EGFP protein (Fig. 5d). Notably, loss of any of these

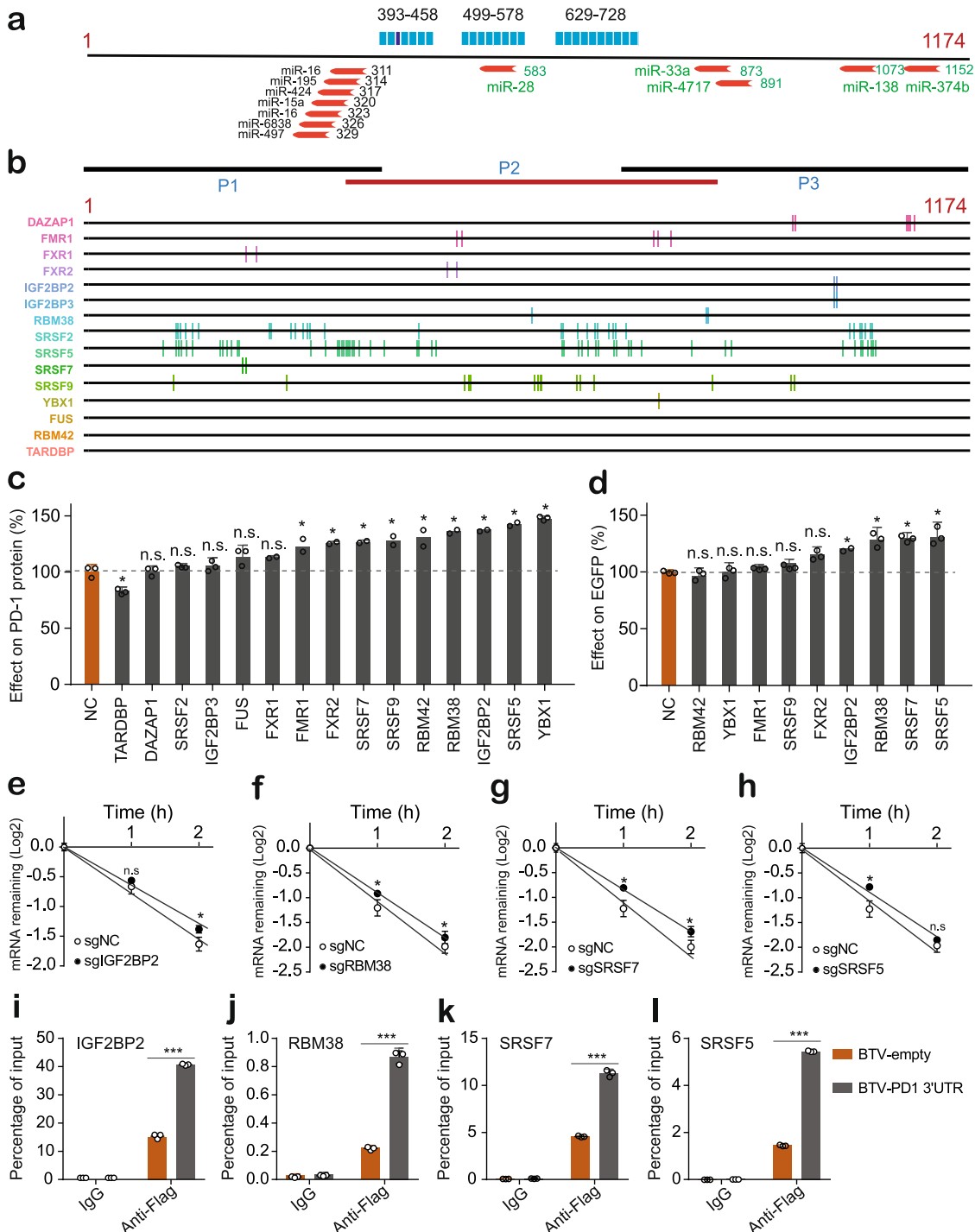

**Fig. 5 The effect of RNA binding proteins on reporter gene expression. a** Positions of known or predicted miRNA binding sites. The green texts represented the miRNA binding sites that have been studied previously; the black ones were predicted miRNA binding sites that are evolutionarily conserved by Targetscan; the blue bars above marked the critical regions responsible for PD-1 3' UTR activity. The numbers right to the red arrows represented the positions of the first nucleotide of the seed sequences of miRNA targets. **b** Position of binding sites (vertical lines) of a portion of predicted RBPs within the PD-1 3' UTR. (**c**) Effect of loss of predicted RBPs on endogenous PD-1 protein; $n = 3$. **d** Effect of loss of predicted RBPs on the regulator activity of the PD-1 3' UTR; $n = 3$. **e–h** Effect of RBPs on PD-1 mRNA stability; $n = 3$. **i–l** RIP-qPCR analysis of PD-1 3' UTR binding with the indicated RBPs. $n = 3$. All values represent mean ± s.d. *, $p < 0.05$ by one-way ANOVA and Dunnett's test (**c–h**); *** $p < 0.001$ by Student's t-test (**i-l**); n.s., no significant difference.

individual RBPs caused only moderate or slight change in the PD-1 expression, which was in consistence with the effect of active regions in the PD-1 3' UTR, further supporting the notion that the inhibitory activity of the PD-1 3' UTR determined by cumulated weak regulation could be beneficial to PD-1 expression

homeostasis. We next asked whether those four RBPs affected mRNA stability. By monitoring endogenous PD-1 mRNA remaining after treatment with Actinomycin D, we found knock-out of each of those RBPs caused slight but significant increase in mRNA stability (Fig. 5e-h), indicating that the PD-1 3' UTR

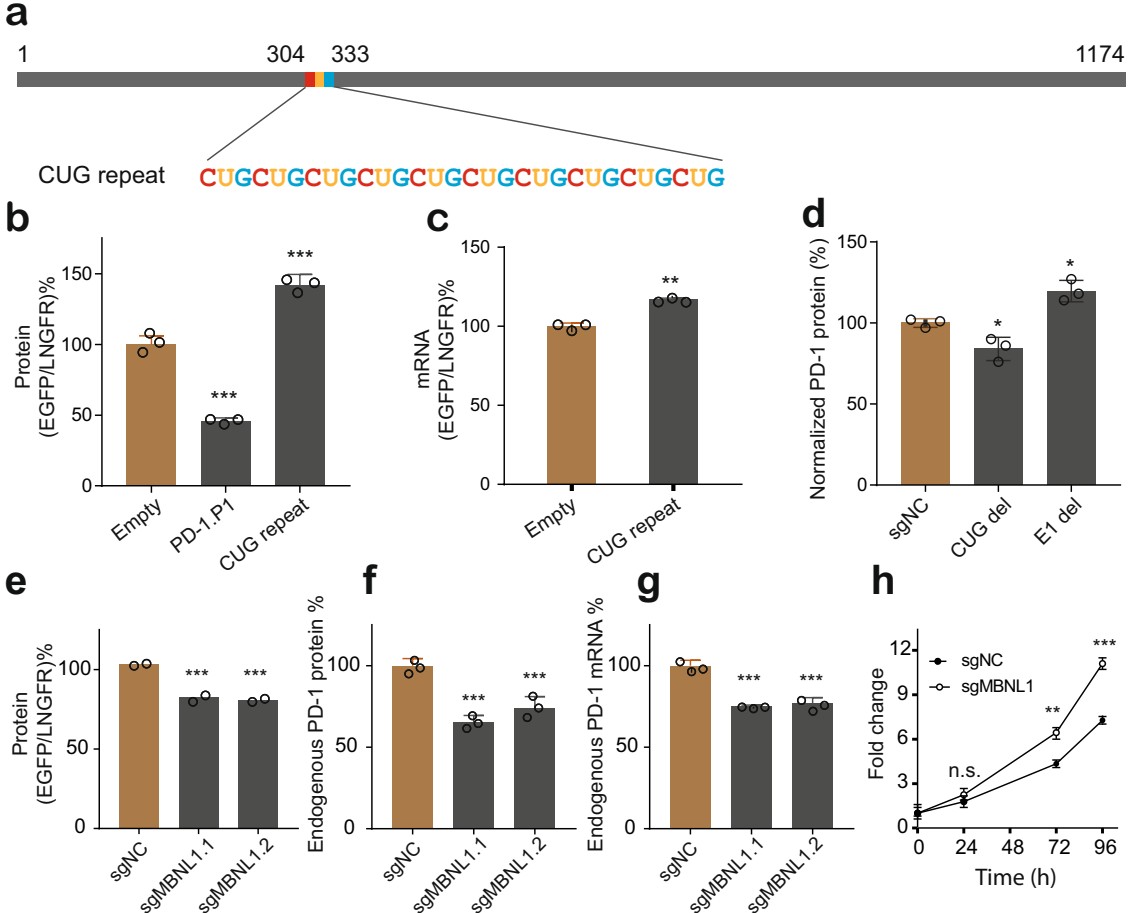

**Fig. 6 MBNL1 binding to the CUG repeat enhances the reporter and endogenous PD-1 expression. a** The position of the CUG repeats in the PD-1 3' UTR. **b, c** The effect of the CUG repeat containing fragment on level of the reporter protein (**b**) and mRNA (**c**); $n = 3$. **d** Effect of endogenously deleting of small active regions on PD-1 expression; CUG del: 283-362; E1 del: 417-464; $n = 3$. **e** Effect of knockout of MBNL1 on the level of reporter protein; $n = 2$. **f, g** Effect of knockout of MBNL1 on endogenous PD-1 protein level (**f**) and mRNA level (**g**); $n = 3$. **h** Effect of knockout of MBNL1 on the proliferation of colon cancer cells; $n = 3$. All values represent mean ± s.d. **, $p < 0.01$, ***, $p < 0.001$ by one-way ANOVA and Dunnett's test (**b, d, e, f**) and Student's t-test (**c, g**); n.s., no significant difference.

mediated mRNA decay was governed by multiple RBPs. To further explore where those RBPs directly bound to PD-1 3'-UTR, we performed RIP-qPCR assay and found that those RBPs were significantly enriched after PD-1 3'-UTR pulldown compared to the control RNA sequence (Fig. 5i-l). We therefore concluded that the PD-1 3' UTR could repress gene expression by binding multiple RBPs that promote mRNA decay.

**MBNL1 interacting with CUG repeats in the 3' UTR enhanced PD-1 expression and could potentially inhibit cancer cell proliferation.** Although the inhibitory activity of the PD-1 3' UTR could be well explained by the cumulated regulation of numerous inhibitory regions, we noticed the fragment P2 exhibited stronger inhibitory activity than the full length of PD-1 3' UTR did (Fig. 3b), implying the PD-1 3' UTR might simultaneously contain enhancing elements somewhere. We therefore scanned the sequence and found a $(CUG)_{10}$ triplet repeat region (Fig. 6a). The CUG repeats are commonly found in premature transcripts as well as in mature mRNAs, including the part of 3' UTR, with the best-known function of modulating efficiency and accuracy of pre-mRNA splicing, mRNA transport, and translation[48]. To examine whether the $(CUG)_{10}$ repeats in the PD-1 3' UTR represented an enhancing element, we tested the

fragment containing the repeats in the BTV reporter. Indeed, it enhanced the reporter protein (Fig. 6b) and steady-state mRNA level (Fig. 6c). Further, we found deletion of the small region containing the CUG repeat (CUG del) using CRISPR-Cas9 expectedly caused modest decrease in PD-1 expression (Fig. 6d). Similarly, deletion of the small region exhibiting repressive activity (E1 del) led to the significant increase in PD-1 expression (Fig. 6d), suggesting there was a good consistence in the results with both reporter assays and endogenous assays.

We next decided to indentify the RBP interacting with the $(CUG)_{10}$ repeat. The well-known CUG-repeat containing sequence is the 3' UTR of *DMPK* mRNA, where the CUG repeat is expanded from normal 5 ~ 37 repeats to mutated 50 ~ 3000 repeats in myotonic dystrophy type 1 (DM1) patients. The expanded CUG repeats bind to and sequester MBNL1 and CUG-BP1 proteins, which are two RBPs required for the alternative splicing of numerous developmentally regulated transcripts[49]. Considering that CUG-BP1 can mediate mRNA rapid decay[50], while MBNL1 has been demonstrated to stabilize transcripts by recognizing GCUU motif[51], we transduced the BTV reporter containing the PD-1 $(CUG)_{10}$ repeat into the cells with the knockout of *MBNL1*. As expected, depletion of MBNL1 significantly reduced the reporter gene expression (Fig. 6e). We further found that knockout of *MBNL1* also reduced endogenous

PD-1 protein and mRNA level (Fig. 6f and g). Therefore, we concluded that MBNL1 could recognize the $(CUG)_{10}$ repeat to enhance PD-1 expression. Since the silencing of PD-1 could promote cancer cell proliferation[52], and we have demonstrated the knockout of MBNL1 reduced the PD-1 expression, we were next wondering whether MBNL1 had effect on cancer cell proliferation by knocking out MBNL1 in WiDr cells. Indeed, depletion of MBNL1 significantly accelerated cell proliferation (Fig. 6h), suggesting that the MBNL1/PD-1 axis could potentially modulate cancer cell proliferation but more direct evidence is required. Taken together, our results demonstrated that the inhibitory and enhancing elements codetermined the regulatory activity of the 3' UTR that governed the PD-1 expression homeostasis.

**PD-1 3' UTRs were functionally conserved across mammals which was coupled with common RBP binding sites or gain/ loss of specific RBP binding sites.** We have demonstrated that the human PD-1 3' UTR played critical roles in the regulation of PD-1 expression. Interestingly, the PD-1 3' UTR sequences are relatively of poor conservation across species compared to the coding regions (Fig. 1a). This inspired us to ask how the PD-1 3' UTRs of other species could affect gene expression, and whether the function and the regulatory model of the human 3' UTR are unique. We first constructed a phylogenic tree based on PD-1 3' UTR sequences of dog, mouse, and twenty primates (including the human) to select several representative 3' UTRs for functional study (Fig. 7a). In specific, the 3' UTRs of human, gibbon, rhesus, green monkey, marmoset, and mouse were selected (Fig. 7a) and tested with the BTV reporter, with the result that all these 3' UTRs substantially decreased reporter gene expression in both WiDr (human) and MC38 (mouse) cells (Fig. 7b), suggesting that the inhibitory activity of the PD-1 3' UTR was conserved during evolution. An explanation could be that most of the RNA binding sites are conserved during the rapid evolution of PD-1 3' UTRs, which was indirectly supported by the observation that the mouse PD-1 3' UTR, without the CUG repeats, exhibited stronger inhibitory activity in both cell types (Fig. 7b). To further support this hypothesis, we used RBPmap to predict and found 68 RBPs that had binding sites shared by the human and the mouse PD-1 3' UTRs (Fig. 7c and Supplementary Figure 1), suggesting that the RBP binding sites were well conserved during evolution despite relatively high sequence divergence. Further, many of these RBPs, including FUBP3 and PUM1 have been demonstrated to control gene expression by binding to the 3' UTRs[53,54]. Notably, the distribution of the RBP binding sites varied between two species, with some overlapped or around the same positions while some localized far away to each other (Fig. 7c). Considering that the mouse and the human PD-1 3' UTRs share many RBP binding sites, we wondered if the mouse PD-1 3' UTR also inhibited gene expression via cumulated weak regulation as did the human one. The mouse PD-1 3' UTR was fragmented into three pieces with some overlap and roughly equal length and tested with the BTV reporter. Indeed, each of the fragments showed only moderate regulatory activity (Fig. 7d), suggesting that the inhibitory activity of the mouse PD-1 3' UTR was the result of many weak regulatory elements or RBP binding sites. Therefore, the PD-1 3' UTRs appeared to evolve to use the same mechanism to maintain PD-1 expression homeostasis, although the pairwise alignment of PD-1 3' UTR sequences between human and mouse showed very low sequence similarity.

We were next interested in whether and how the sequence divergence affects the regulatory activity of the PD-1 3' UTRs during evolution. By comparing PD-1 3' UTRs among primates, we observed 14 indels with the length greater than 5 nucleotides.

Among them two variations attracted our attention. The first one was the number of CUG repeats. The human PD-1 3' UTR contains 10 CUG repeats (Fig. 6a, Fig. 7e), which was able to increase mRNA stability dependent on MBNL1 as we showed above (Fig. 6e-g). However, we found fewer numbers of CUG repeats among other primates, 8 for the chimp, 7 for the orangutan, and 6 for the crab-eating macaque, for example (Fig. 7e). Presumably, fewer number of CUG repeats bind less of MBNL protein, thus consequently decreasing gene expression by destabilizing mRNA transcript. To test this presumption, we examined four 3' UTR fragments containing 6, 7, 8, or 10 CUG repeats mentioned above with the reporter. As expected, fewer CUG repeats tended to cause less gene expression in both human and mouse cells (Fig. 7f). Notably, no such CUGx repeats (x >= 3) exist in mouse and dog PD-1 3' UTR at all, implying the possibility that greater number of CUG repeats were gained to stabilize PD-1 mRNA transcript during primate evolution. Second, there was a region that was lost in Hominoidea, including the human and the chimp but reserved in other primates (Fig. 7g). To examine whether this region was active or not, we tested a short fragment containing this region from the rhesus (referred to as 'rhesus Elmt') in the reporter, with the result that the rhesus Elmt decreased the reporter protein expression significantly in both cell types (Fig. 7h), suggesting a potential inhibitory element. Interestingly, the rhesus PD-1 3' UTR contains a shorter stabilizing (CUG)x portion as well as the destabilizing Rhe Elmt compared with the human PD-1 3' UTR (Fig. 7e and g), but we did not see the rhesus 3' UTR to be more destabilizing compared to the human UTR (Fig. 7b). One explanation could be that because the (CUG)x and the Rhe Elmt are not the only elements that affect mRNA stability, there could be other destabilizing elements that the human PD-1 3' UTR contains but the rhesus one does not. Taken together, it appears that the PD-1 3' UTR evolved to adapt to cellular contexts with a mechanism via gain or loss of functional elements.

## Discussion

PD-1 is a multifunctional molecule with the primary role in induction and maintenance of T cell peripheral tolerance. PD-1 deficient mice exhibited a stronger response to IgM stimulation and spontaneously developed autoimmune disorders, such as lupus-like glomerulonephritis[4]. In contrast, highly expressed PD-1 on exhausted T cells in patients is associated with cancer or chronic viral infection. Moreover, there is particularly high expression of PD-1 on intratumoral NK cells, which results in reduced degranulation and reduction in the cytotoxic functions of NK cells[5,6]. Such pathological elevation of PD-1 is one of the major causes for T cell exhaustion. Antibodies have been successfully used to treat human cancers by blocking the PD-1 signaling to enhance T cell activity and augment the lytic function of NK cells, illustrating the great value of PD-1 as a target for cancer therapy[55]. However, despite the benefits, PD-1 antibodies were not sensitive in most of cancer patients. Therefore, it has become a fundamental need to unveil the genetic information that is involved in the regulation of PD-1 expression and how different genetic information orchestrates to maintain PD-1 expression homeostasis. Knowledge from these studies could potentially facilitate elucidating the molecular events underlying T-cell exhaustion and developing the next generation of drugs for cancer therapy.

In this report, we systematically investigated the roles for the PD-1 3' UTR in the control of gene expression. Quantification of both reporter gene and endogenous gene expression demonstrated that the PD-1 3' UTR could dramatically reduce protein level and steady-state mRNA level, which was tightly coupled with rapid mRNA decay. Whereas many groups have previously

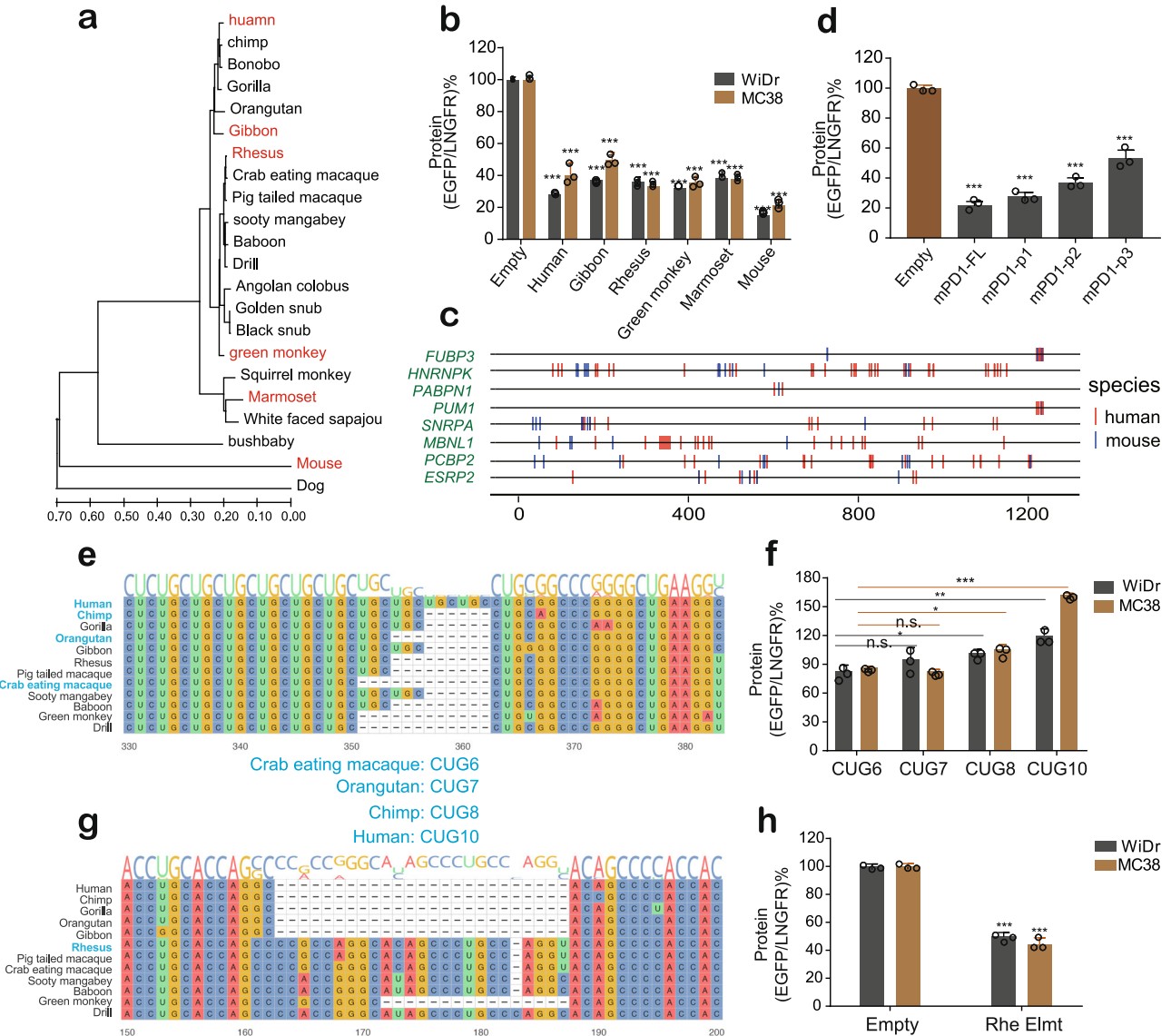

**Fig. 7 Evolutionary analysis and functional validation of the PD-1 3′ UTR. a** The evolutionary tree based on the PD-1 3′ UTR sequences of the mouse and twenty primates (including the human). **b** Quantification of the effect of the 3′ UTRs derived from human, gibbon, rhesus, green monkey, marmoset, and mouse; *n* = 3. **c** Distribution of the positions of predicted RBP binding sites conserved between the human and the mouse PD-1 3′ UTRs. Eight representative RBPs were selected to display their predicted binding sites, either overlapped or separated; the rest were displayed in the Supplementary Figure 1. **d** Measurement of shorter fragments of the mouse PD-1 3′ UTR on reporter gene; *n* = 3. **e** Alignment of partial primate 3′ UTRs to show the varieties of CUG repeats. **f** Evaluation of the effect of varieties of CUG repeats on reporter gene expression; *n* = 3. **g** Alignment of partial primate 3′ UTRs to show loss of functional sequences during evolution. **h** Evaluation of the effect of the lost region on reporter gene expression; *n* = 3. All values represent mean ± s.d. **, *p* < 0.01, ***, *p* < 0.001 by one-way ANOVA and Dunnett's test (**b**, **d**, **e**, **f**) and Student's t-test (**c**, **g**); n.s., no significant difference.

reported several types of regulators of PD-1, including transcription factors, epigenetic modification, miRNAs, and polyubiquitination, this work revealed that the PD-1 3′ UTR could function as a negative mediator for repression of PD-1 expression. Since the PD-1 3′ UTR can directly affect PD-1 expression, cell phenotype, and biological function, we would think the post-transcriptional regulation mediated by PD-1 3′ UTR could be as important as transcriptional or other types of regulation. This finding is quite important because it could help explain how PD-1 expression is kept at an appropriate level to maintain the balance between immune tolerance and exhaustion. Since high dosage of PD-1 is one of the major causes for cancer, it is likely that the highly repressive PD-1 3′ UTR may play a crucial role in the restriction of cancer immune escape. Interestingly, it has been demonstrated that truncation of the 3′ UTR of PD-L1 (the ligand

for PD-1) lead to a marked elevation of aberrant PD-L1 transcripts by stabilizing mRNA in multiple common human cancer types. Disruption of the Pd-l1 3′ UTR in mice enables immune evasion of EG7-OVA tumor cells with elevated Pd-l1 expression in vivo, indicating similar roles of the 3′ UTRs in the regulation of PD-1 and PD-L1 expression[56].

To identify the functional elements responsible for the regulatory activity of the PD-1 3′ UTR, we initially examined the sequence but did not find any AREs, GREs, or CDEs except several miRNA targets such as miR-16-5p, miR-195-5p, and miR-424-5p. We therefore employed experimental approaches to localize regulatory elements by using fragmentation, truncation, and mutation. Unexpectedly, it appeared that the robust regulatory activity of the PD-1 3′ UTR was attributed to many weak regulatory regions rather than one or few dominant ones. It is

common that sequence variation and mutagenesis within a regulatory region could largely disrupt its activity. For example, variation within the 3' UTR of *HLA-C* regulates the binding of hsa-miR-148 to its target site, resulting in relatively low surface expression of alleles that bind this miRNA and high expression of *HLA-C* alleles that escape post-transcriptional regulation[57]. We therefore proposed that the regulatory activity governed by multiple weak elements could do better to maintain PD-1 expression homeostasis than one or few dominant ones, as any variation or mutagenesis occurred within the strong element that governs the regulatory activity of a 3' UTR could lead to loss of the entire activity. In contrast, mutagenesis occurred in one of the many weak regulatory elements would cause only subtle effect on the whole 3' UTR (Fig. 4g). Further, the cumulation of weak regulation might represent a general model how gene expression is controlled by 3' UTRs, since a 3' UTR typically contains several, decades, or hundreds of RBP binding sites dependent on length and nucleotide composition. We further found that the regulatory activity of the PD-1 3' UTR relied on multiple RBPs by binding to the 3' UTR, implying that the RBPs and the 3' UTR elements orchestrate to maintain PD-1 expression homeostasis.

Interestingly, the PD-1 3' UTR evolved much rapidly than most of 3' UTRs of other genes. It could be a general explanation that the PD-1 3' UTRs rapidly evolved to meet the need of different species. However, our work revealed that the PD-1 3' UTRs of the mouse and several other primates were functionally similar by repressing gene expression, which was quite surprising but actually highlighted the conserved function of the PD-1 3' UTRs in the balance of PD-1 expression in the immune systems during evolution. Since the expression of the PD-1 is required to be maintained at a low level in T cells to prevent T cell exhaustion in both the human and the mouse[2,9], it seems necessary to conserve the inhibitory activity of the PD-1 3' UTR. We further found that the conserved function of the PD-1 3' UTR was associated with many common RBP binding sites, which could potentially interact with the conserved RBPs to co-repress the PD-1 expression. On the other hand, a specific 3' UTR could differentially affect gene expression in different species. For example, the human and the mouse PD-1 3' UTRs repressed gene expression more in the human cells than in the mouse cells (Fig. 7b), suggesting the RBP expression profile vary across species, thus an RBP binding site that works in one species might become useless in other species due to lack of relevant RBPs. To adapt to such cellular context change, the PD-1 3' UTR evolved to selectively conserve or lose some functional regions such as the Rhe Elmt in specific lineages (Fig. 7g and h). In the meanwhile, the PD-1 3' UTR could gain more RBP binding sites during evolution to fine-tune the PD-1 expression by utilizing specific RBPs. For example, the human PD-1 3' UTR gained more CUG repeats that bound to MBNL1 to improve mRNA stability (Fig. 7e and f). Therefore, our work illustrated how the PD-1 3' UTRs used strategies of evolution to modulate gene expression in mammals.

## Methods

**Subcloning of 3' UTRs into Reporters.** To test the effects of 3' UTRs on reporter genes in human cells, we used the previously developed plasmid (BTV)[29]; the BPV was used to test 3' UTRs in mouse cells[58]. The PD-1 3' UTR and the fragments derived from the PD-1 3' UTR were amplified from BEAS-2B and primate genomic DNA (kindly offered by Dr. Bing Su, Kunming Institute of Zoology, CAS, China) and ligated into the BTV after digestion with MluI and SbfI (or PacI). All inserts were verified by sequencing. All oligo DNA sequences used in this work were list in Supplementary Data 2.

**Cell Culture.** BEAS-2B, HelaS3, WiDr cells stably expressing the tTA tetracycline transactivator, MC38, and HEK-293T cells were maintained in DMEM with 10% FBS, 2 mM glutamine, 0.1 mM nonessential amino acids; Jurkat cells stably

expressing the tTA tetracycline transactivator and MOLT-4 cells, not registered in ICLAC, were maintained in RPMI 1640 with 10% FBS, 2 mM glutamine, 0.1 mM nonessential amino acids. Jurkat, BEAS-2B, HelaS3, WiDr cells stably expressing the tTA tetracycline transactivator, and HEK-293T were obtained from David Erle lab, University of California, San Francisco; MC38 cells was a gift from Dr. Xiaojun Xia, Sun Yat-sen University Cancer Center; MOLT-4 cells was a gift from Dr. Yangqiu Li, Jinan University, Guangzhou. BEAS-2B, HelaS3, WiDr cells with tTA and HEK-293T cells were tested to ensure no mycoplasma contamination by PCR method with a conditioned medium as a template using the primer pair: GGGAGCAAACAGGATTAGATACCCT, TGCACCATCTGTCACTCTGTTAA CCTC; mycoplasma positive MOLT-4 cells were treated with Plasmocin™ (InvivoGen) until clearance. We did not authenticate any cell lines.

**Lentivirus production and transductions.** To make lentivirus, HEK293T cells (0.6 M per well) were prepared in 1 ml of medium in 12-well plates one day before. By the time of transfection, 600 ng of backbone plasmid, 400 ng of PAX2 plasmid, and 200 ng of VSV-G plasmid were mixed with 50 ul of OptiMEM to form Mix I; 3.6ul of PEI (1ug/ul, Polysciences, 24765-1) was mixed with 50ul OptiMEM to form Mix II. The Mix I and Mix II were then combined and incubated for 20 min at room temperature; the final mixture was added to the HEK 293 T cells. Conditioned medium containing lentivirus was collected 48 h posttransfection and used immediately to infect cells or frozen at −80 °C. To transduce cells, we diluted conditional medium with fresh medium (2:3) and added to the cells. After 24 h growth, the cells were replaced with fresh medium. Transduced cells were allowed to grow for 72 h before flow cytometry analysis.

**Cell staining and flow cytometry.** To detect endogenously expressed PD-1 protein on the surface of MOLT-4 cells, we spun down 50 K cells, followed by staining with 50 ul APC conjugated anti-human CD279 (1:100 diluted, Biolegend, 367406) in PBS containing 10% FBS on ice for 30 min. After twice of wash with PBS, cells were resuspended in 100ul PBS and fixed with 1% paraformaldehyde. 20,000 events were collected by running the stained cells on the flow cytometer (Beckman Cytoflex). The PD-1-positive cells were gated based on unstained cells. To detect BTV reporter gene expression, 10-20 K cells as indicated in the main text were plated in 96-well plates one day before adding lentivirus. The medium was replaced with fresh medium 24 h after adding virus. Cells were allowed to grow for additional 48 h before harvesting for staining with Alexa 647-conjugated anti-LNGFR Ab (Biolegend, L345114, 100 ug/ml), 1:100 diluted in PBS containing 10% FBS, for 30 min on ice. After twice of wash with PBS, cells were resuspended in 100ul PBS and fixed with 1% paraformaldehyde. 20,000 events were collected by running the stained cells on the flow cytometer (Beckman Cytoflex). To determine the activity of 3' UTRs, we collected antibody-stained but untransduced cells as a negative control to gate the population of LNGFR-positive cells; the median fluorescence intensity (MFI) of EGFP and LNGFR was calculated for the positive cells; the MFI ratio of EGFP/LNGFR represented the activity of 3' UTRs.

**Quantitative real-time PCR (qRT-PCR).** The Total RNA Kit I (Omega Bio-tek, R6834) was used to extract cellular total RNA. DNase I was used to eliminate possible DNA contamination in RNA samples before synthesis of cDNA with the PrimeScript RT reagent Kit with gDNA Eraser (Takara, RR047A). The TB Green Premix Ex Taq (Takara, RR420A) was used to detect target abundance on Applied Biosystems QuantStudio 5 Real-Time PCR Systems. *LNGFR* was used to normalize the steady state mRNA level of the reporter gene *EGFP*; *GAPDH* was used for normalization of reporter mRNA in decay assays; *GAPDH* was used to normalize all endogenous mRNA level. Relative RNA levels were evaluated using ΔΔCt method.

**Monitoring RNA degradation.** To monitor endogenous mRNA decay, 0.75 M cells in 1 ml medium in 12-well plates were treated with actinomycin D (5 ug/ml, final) and harvested for total RNA extraction at time 0, 1, 2, 4 h. To monitor reporter mRNA decay, transduced cells were propagated and divided equally into multiple wells in 12-well plates one day before adding doxycycline solution (1 μg/ml, final) to the medium. Cells were harvested 0, 1, 2, 4, or 8 h after stopping transcription for total RNA extraction. The target mRNAs remaining at each time point was measured with qRT-PCR as stated above; mRNA decay rates were presented by exponential fitting.

**CRISPR-Cas9 knockout assays.** To knock out protein-coding genes, sgRNAs were designed with the online tool CRISPick and cloned in the vector Lenti-CRISPRv2 (Addgene # 52961), followed by packaging into lentivirus, which was then used to infect cells as indicated. 48 h post-infection, 1 ug/ml of puromycin was used to screen positive cells. After propagation, the portion of cells were collected to extract genomic DNA, followed by PCR amplification for analysis of knockout efficiency with TIDE[59]. Multiple gRNAs were designed for each targeting site, and only the ones with KO efficiency greater than 70% were used for subsequent experiments. To knock out the DNA sequence encoding the 3' UTR fragment, two gRNAs respectively targeting the proximal and the distal terminals were cloned in LentiCRISPRv2 and the modified LentiCRISPRv2 (blasticidin resistant), followed by co-infection in the cells. After screening with puromycin and blasticidin, the

portion of cells were collected to extract genomic DNA, followed by PCR amplification for analysis of knockout efficiency in agarose gel.

**RNA immunoprecipitation (RIP)–qPCR analysis**. The previous protocol was followed with some modification[60]. Briefly, 293 T cells in 10-cm dishes with a confluence of 50-60% were transfected with BTV reporter, the tTA-expressing construct, and the 3xFlag tagged RBP-expressing construct (5ug each). 24 h post-transfection, cells were harvested in 200 µl RIP Lysis Buffer (10 mM HEPEs, pH7.0, 100 mM KCl, 5 mM MgCl2, 0.5% Nonidet P40, additional 1 mM DTT, 400 µM RVC and 0.5% PI added immediately before use). 50 µL magnetic beads were mixed with 4 µL anti-Flag (ABclonal, AE092, 1 mg/ml) and anti-rabbit IgG (ABclonal, AC005, 1 mg/ml) before the addition of cell lysates. Then incubate in an inversion mixer for 4 °C overnight. After the treatment of TRK Buffer andnext with proteinaseK, interested RNAs were eluted from immunoprecipitated complex and purified for further analysis using qPCR. Relative enrichment was normalized to the input: %Input $= 2^{\text{Ct [IP] – Ct [input-3.3]}}$.

**Cell proliferation assays**. $2 \times 10^4$ cells were prepared in 96-well plates containing 100ul medium per well. 0, 24, 72 and 96 h later, the medium was replaced with 100 µl fresh medium containing 10 µl of CCK reagent (TransDetect Cell Counting Kit, TransGen) and mixed. After 3 h' incubation at 37 °C, the absorbance was read with a microplate reader (Synergy H1; Bio-Tek, Winooski, USA) at 450 nm.

**Analysis of 3' UTR conservation**. UCSC phastCons conservations scores for human genome (hg38) across 99 vertebrate species were retrieved from R package phast-Cons100way.UCSC.hg38. (https://bioconductor.org/packages/phastCons100way.UCSC.hg38/). The genomic coordinates of CDSes and 3'UTRs for all genes were retrieved from R package EnsDb.Hsapiens.v86. Genes located on the nonstandard chromosomes were removed, and in total 21,050 genes were kept in our analysis. The average phastCons scores of CDSes and 3'UTRs for each gene were calculated with R package Genomic-Scores. The phastCons is a measurement of evolutionary conservation at base level and was estimated using a hidden Markov model based on multiple species alignment.

**Statistics and Reproducibility**. Statistical analyses were performed using GraphPad Prism 7. Sample sizes were determined based on previous studies using similar experiments. Individual information about the sample size and statistical tests is described in the figure legends. All data are presented as the Mean ± SD of at least three independent tests. Statistical analysis was performed using one-way ANOVA, Dunnett's test, and Student's t-test as indicated in the caption. $p < 0.05$ was considered statistically significant. All experiments were performed at least twice with similar results.

**Reporting summary**. Further information on research design is available in the Nature Portfolio Reporting Summary linked to this article.

## Data availability

The original data used in this work are available from the authors upon request. Numerical source data for the Fig. 1a can be found in Supplementary Data 1; numerical source data for the Figs. 1d-f, 2c-j, 3b-d, 3f-g, 4b-f, 4h, 5c-l, 6b-h, 7b, 7d, 7f, and 7h can be found in Supplementary Data 3. The original uncropped and unedited PCR gel image can be found in Supplementary Figure 2.

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

## Acknowledgements

This work was supported by the National Natural Science Foundation of China (31971224), the Natural Science Foundation of Guangdong Province (2021A1515012125), the Natural Science Foundation of Shenzhen (JCYJ20190807162007540), and the Zhujiang Talent Program-Young Scholar (2019QN01Y284).

## Author contributions

W.Z. conceived of key aspects of the project, designed the experiments, and wrote the manuscript. X.Lai, R.L., P.W., M.L. and C.X. carried out the experimental work and analyzed the data; X.Li, X.Lai, and R.L. edited the manuscript. X.Li and Q.C. performed evolution analysis and prediction of RBP binding sites.

## Competing interests

The authors declare no competing interests.
