## [Peer Review File · Communications Biology]

Reviewers' comments:

Reviewer #1 (Remarks to the Author):

The manuscript "Cumulation of Weak Regulation Mediated by the Repressive 3' UTR Facilitates the Maintenance of PD-1 Expression Homeostasis in Mammals" by Lai et al. dissects the functional elements of the PD-1 3' UTR in mRNA decay. The interesting finding that multiple weak cis-regulatory elements jointly control mRNA stability and buffer against mutational effects is compellingly supported by extensive mutational analyses. However, the manuscript does not provide information on the biological relevance of this regulation and/or its mechanistic basis, which limits the overall impact of the study and the interest to the broader research community.

Major comments:

1) What is the biological significance of PD-1 mRNA instability? Is the 3'-UTR effect more important, equally important, or less important than transcriptional regulation of PD-1? Do UTR knock-out cells (as shown in Figure 2) exhibit a phenotype in immune regulation?

2) Please provide more evidence for your claim that PD-1 mRNA decay is mediated by multiple RNA-binding proteins. Of particular concern is that only 4 of 9 tested RBPs affect reporter gene regulation via the 3'-UTR. In contrast, 11 of 15 RBPs affect the endogenous PD-1 mRNA, even two of the three controls. Overall, this analysis confirms that PD-1 expression can be easily perturbed, but does not directly support RBP-dependent regulation via the 3'-UTR. Please provide direct evidence for RBP interaction with the 3'-UTR of PD-1.

Further questions regarding regulation by RBPs:

- On what basis were the 12 RBPs in Figure 5 selected from the 86 predicted?

- Are the binding sites for these 12 RBPs not conserved in mice? On what basis were the 8 RBPs in Figure 7 selected?

- What about miRNA-dependent regulation? Do the binding sites of the known PD-1 regulating miRNAs overlap with the critical regions for mRNA decay identified in the manuscript?

3) The authors mention in the introduction that they compared the conservation of "21,050 genes from 99 vertebrates" and found that the 3'-UTR of PD-1 evolves rapidly compared to the coding region. a) Please provide the data from this analysis. b) Is this not to be expected? Non-coding regions evolving faster than coding regions? Please explain the important point here.

Minor/technical comments:

Figure 1

c: What is the function of the red line shown in the dot plots?

d: Please explain the method used to determine the ratio of EGFP/LNGFR. Did all events shown in c contribute to the mean fluorescence intensity reported in the Methods section?

f: Please change the y-axis to a logarithmic axis and calculate the half-life of the mRNAs with and without the 3'-UTR of PD-1.

Figure 2

a: PCR detection after deletion of the 3'-UTR using CRISPR-Cas9 shows two bands. Thus, the knockout is not homozygous. Please comment on this in the main text and how this affects the results/interpretation of the data shown.

e: Please change the y-axis to logarithmic and calculate the half-life of PD-1 mRNAs with and without the full 3'-UTR. Please add error bars.

Figure 3

Please indicate the localization in the 3'-UTR using the nucleotide position within the full 3'-UTR not the P2 fragment throughout the study to avoid confusion with numbering in other figures (e.g., Figure 6).

d: The experiment/figure does not contribute to the manuscript. Either comment on the relevance of the figure or remove it from the manuscript.

g and h: Exchange the position of g and h to match the data presented previously.

Figure 5

a: Please indicate the positions of P1, P2, and P3 in the scheme.

b and c: How efficient is the knockout for the different RBPs targeted by CRISPR-Cas9 (Methods section: TIDE)?

Figure 6

g: MBLN1 knockout increases proliferation in colon carcinoma cells. Is this related to PD-1 regulation by MBNL1? Please comment on the relevance to the main topic of the paper.

Figure 7

b and e-h: The rhesus PD-1 3'-UTR contains a shorter stabilizing (CUG)_x portion as well as the destabilizing Rhe Elmt compared with the human PD-1 3'-UTR. Therefore, one would expect the rhesus UTR to be more destabilizing compared to the human UTR. This is not the case (according to b). Please comment.

Reviewer #2 (Remarks to the Author):

In this study, Lai et al. use reporter assays and endogenous gene editing to elucidate the regulatory role of the 3' UTR of the gene encoding PD-1 (PDCD1) using sequences from human, mouse and several non-human primates. These studies reveal that while the full ~1kb 3' UTR sequence of PD-1 has a large regulatory effect that reduces expression of a reporter construct or the endogenous transcript, that this regulatory activity is distributed over a number of regions across the sequence, each of which has a relatively small effect in isolation. Additionally, they investigate the activity of a number of RNA binding protein genes that act in trans to regulate PD-1 expression level, and similarly discovered that no single factor has a dominant effect, with a number of factors having a subtle effect, suggesting a coordinate regulation of gene expression through the small contributions of a number of RNA binding proteins.

Overall, these findings are in line with our understanding of post-transcriptional regulation of gene expression, and form a valuable example of deep characterization of the contributions of individual regulatory elements in a single 3' untranslated region to work as an ensemble to regulate gene expression. Additionally, the fact that their experimental approach is informed by an evolutionary perspective adds value to their findings: the lower level of conservation of the 3' UTR relative to the coding sequence that the authors note in Figure 1 is a general characteristic of many 3' UTRs. Their observation of apparent functional activity despite an overall lower level of sequence conservation, and particularly their dissection of the contribution of variation in 3' UTR sequence in PDCD1 across mammals to regulatory activity (Figure 7) is appreciated. Finally, their choice of PD-1 as an example UTR to study in detail is apt. This gene is an important therapeutic target in its own right, and genetic variation altering its exact expression level may have important biological consequences, in line with the authors' hypothesis that the 3' UTR of PD-1 may be a locus undergoing rapid evolution that drives its sequence variation across species.

The study is well conducted and contains important controls that support the authors' main findings and interpretations and will be of value to immunologists and biologists interested in understanding post-transcriptional control of gene expression generally, or researchers seeking to conduct similar studies on their own gene of interest. As such, it merits publication in *Communications Biology*, but would be strengthened by a small number of focused experiments and some key revisions to the discussion:

In particular, the studies in Figures 2 and 3 suggest that while the endogenous 3' UTR has a moderately strong effect (~60% increase in PD-1 protein expression with the whole UTR KO in Fig. 2C), that large truncations had a strong regulatory effect in a reporter assay (Fig. 3C). Then, the data with small substitution mutations of individual regulatory elements had relatively small effect size. Therefore, it would be of interest to know what the effect of smaller individual deletions would be in regulating the endogenous expression of PDCD1 in a cell type that expresses this gene natively. This could be done, for example, using the 3' UTR dissection method that the senior author of this

manuscript published <https://www.ncbi.nlm.nih.gov/pmc/articles/PMC5737544/>

Additionally, it would strengthen the paper to demonstrate that the regulation dissected throughout the manuscript has a biological impact on some measurable aspect of lymphocyte function (in donor T cells or Jurkat cells, for example) in line with the authors' hypothesis that 3' UTR sequence variation can alter immunological outcomes to an extent that it drives natural selection acting upon this locus. Finally, in their discussion of the evolution of PD-1 the authors sometimes use language that does not totally accurately capture the current understanding of PD-1 and its role in tumor biology. For example the claim that "pathological elevation of PD-1 is one of the major causes for cancer immune evasion" conflates T cell exhaustion with immune evasion. A careful examination of the claims made in this section with the current state of the field is warranted.

Re: Revision of COMMSBIO-22-3584-T

Point-by-point response to reviewers' comments:

Reviewer #1 (Remarks to the Author):

The manuscript "Cumulation of Weak Regulation Mediated by the Repressive 3' UTR Facilitates the Maintenance of PD-1 Expression Homeostasis in Mammals" by Lai et al. dissects the functional elements of the PD-1 3' UTR in mRNA decay. The interesting finding that multiple weak cis-regulatory elements jointly control mRNA stability and buffer against mutational effects is compellingly supported by extensive mutational analyses. However, the manuscript does not provide information on the biological relevance of this regulation and/or its mechanistic basis, which limits the overall impact of the study and the interest to the broader research community.

Major comments:

R1.C1: What is the biological significance of PD-1 mRNA instability? Is the 3'-UTR effect more important, equally important, or less important than transcriptional regulation of PD-1? Do UTR knock-out cells (as shown in Figure 2) exhibit a phenotype in immune regulation?

Reply: To answer the 1st and the 3rd questions, we examined the activation status of the MOLT-4 cells with deletion of the 3' UTR by measuring the expression of specific molecular markers. We found that the T cell activity was inhibited in the knock-out cells compared to the cells with an intact 3' UTR. We also measured Ki67 expression and performed CCK 8 assays to detect the rate of MOLT-4 cell proliferation. It was showed that the cells with loss of the PD-1 3' UTR proliferated much faster than control cells. These results allowed us to conclude that knock-out of PD-1 3' UTR in MOLT-4 cells could alter cell phenotypes including more rapid cell proliferation and less T cell activity. Therefore, the mRNA instability mediated by PD-1 3' UTR is biologically significant at least by participating in regulation of T cell activation and cell proliferation. We added this new result to Fig. 2f-i and in the text (see p.10 line22 - p.11 line7).

Since the 3' UTR is able to directly affect PD-1 expression, cell phenotype, and biological function, we would think the post-transcriptional regulation mediated by PD-1 3' UTR is equally important to transcriptional regulation. We added this information in Discussion (see p.28 lines 3-6).

R1.C2: Please provide more evidence for your claim that PD-1 mRNA decay is mediated by multiple RNA-binding proteins. Of particular concern is that only 4 of 9 tested RBPs affect reporter gene regulation via the 3'-UTR. In contrast, 11 of 15 RBPs affect the endogenous PD-1 mRNA, even two of the three controls. Overall, this analysis confirms that PD-1 expression can be easily perturbed, but does not directly support RBP-dependent regulation via the 3'-UTR. Please provide direct evidence for RBP interaction with the 3'-UTR of PD-1.

Reply: To test whether PD-1 mRNA decay is mediated by multiple RNA-binding proteins, we measured the decay rate of PD-1 transcript before and after knockout of four RBPs: IGF2BP2, RBM38, SRSF7, and SRSF4, which affected reporter gene expression via the 3'-UTR. Indeed, knockout of each of these RBPs caused slight but significant increase in mRNA stability (see p.19 lines 15-18, and Fig. 5e-h), indicating that the PD-1 mRNA decay is mediated by multiple RBPs.

To further provide direct evidence for RBP interaction with the PD-1 3'-UTR, we performed RIP-qPCR to explore the interaction between these four RBPs and the PD-1 3'-UTR. Results showed that each of these RBPs was significantly enriched after PD-1 3'-UTR pulldown compared to the control, demonstrating direct interaction between these RBPs and the PD-1 3'-UTR. We added this new result to Fig. 5i-l and in the text (see p.19 line 18-22).

Further questions regarding regulation by RBPs:

R1.C2a: On what basis were the 12 RBPs in Figure 5 selected from the 86 predicted?

Reply: These proteins were selected based on their previously known function particularly in control of mRNA stability. In specific, “DAZAP1 is a component of complexes that are crucial for the degradation and silencing of mRNA; FMR1 can bind 3' UTRs to contributes to maternal RNA degradation; FXR1 and FXR2 are involved in the transport, translation, and degradation of mRNA; YBX1 can decrease mRNA stability of Pink1 and Prkn; IGF2BP2/3 can modulate mRNA stability and translation; Rbm38 is required for p63 mRNA degradation; members of the SRSF family such as SRSF3 play roles in splicing and regulate additional aspects of RNA metabolism like alternative polyadenylation, mRNA export”. We added this in the main text (see p. 18 lines 17 – 24).

R1.C2b: Are the binding sites for these 12 RBPs not conserved in mice? On what basis were the 8 RBPs in Figure 7 selected?

Reply: 8 out of 12 are conserved, which are FMR1, FXR2, IGF2BP2, IGF2BP3, RBM38, SRSF2, SRSF5, SRSF9; we added this information in the main text (see p. 18 line 24 – p.19 line2).

The 8 RBPs shown in Fig. 7c were selected to display representative distribution of the positions of predicted RBP binding sites conserved between the human and the mouse PD-1 3' UTRs, which were either overlapped or separated; the rest were displayed in the Supplemental Fig S1. We added this in the Figure 7c legend (see p. 23 lines 7-10).

R1.C2c: What about miRNA-dependent regulation? Do the binding sites of the known PD-1 regulating miRNAs overlap with the critical regions for mRNA decay identified in the manuscript?

Reply: As mentioned in the Introduction section, several miRNAs have been shown to repress gene expression by binding to the PD-1 3' UTR, including miR-4717, miR-374b, miR-28, miR-138, miR-33a. We additionally analyzed by Targetscan and found seven evolutionarily conserved binding sites that are predicted to bind miR-16, miR-195, miR-424, miR-15a, miR-16, miR-6838, miR-497 or other family members. To see whether these miRNA targeting sites overlap with the critical regions for mRNA decay, we added a new panel (a) to Figure 5, plus the legend (p. 17 lines3-8), from which we can clearly see that only miR-28 overlaps with the critical regions. We added this information in the main text (p. 18 lines8-13).

R1.C3: The authors mention in the introduction that they compared the conservation of "21,050 genes from 99 vertebrates" and found that the 3'-UTR of PD-1 evolves rapidly compared to the coding region. a) Please provide the data from this analysis. b) Is this not to be expected? Non-coding regions evolving faster than coding regions? Please explain the important point here.

Reply:

a) We provided the data by storing in the file Supplemental Table 1, and also mentioned it in the main text (p. 8 line 5).

b) We apologize for the confusion. Non-coding regions including 3' UTRs generally evolve faster than coding regions. Here we didn't want to compare the evolutionary rate between the CDS and the 3' UTR of PD-1, but wanted to compare the evolutionary rate of the PD-1 3' UTR with 3' UTRs of other genes. Our results showed that among 21,050 protein-coding genes, the conservation score of PD-1 CDS ranked at the position of 11621 (top 55%), but that of the PD-1 3' UTR ranked at 19609 (top 93%), demonstrated that the PD-1 3' UTR evolves much faster than most 3' UTRs of other genes. We fixed in the main text (p. 8 lines 2-7).

Minor/technical comments:

R1.mC-Figure 1

c: What is the function of the red line shown in the dot plots?

Reply: We drew a red line in order for easier observation of the difference between the control and the PD-1 3'-UTR, which was sort of redundant, so we removed it (see Fig. 1c).

d: Please explain the method used to determine the ratio of EGFP/LNGFR. Did all events shown in c contribute to the mean fluorescence intensity reported in the Methods section?

Reply: To determine the activity of 3' UTRs (the ratio of EGFP/LNGFR), we collected antibody-stained but untransduced cells as a negative control to gate the population of LNGFR-positive cells (the population right to the vertical dotted line on Fig.1 panel c); the median fluorescence intensity (MFI) of EGFP and LNGFR was calculated for the positive cells (see p. 7 line11); the MFI ratio of EGFP/LNGFR represented the activity of 3' UTRs. We added this information in the Methods section (see p. 33 lines8-11) and in the Fig. 1c legend (see p. 7 lines 14-15).

f: Please change the y-axis to a logarithmic axis and calculate the half-life of the mRNAs with and without the 3'-UTR of PD-1.

Reply: Done. See p. 9 line2 and Fig.1f.

R1. mC -Figure 2

a: PCR detection after deletion of the 3'-UTR using CRISPR-Cas9 shows two bands. Thus, the knockout is not homozygous. Please comment on this in the main text and how this affects the results/interpretation of the data shown.

Reply: We added the statement that "To be emphasized, we failed to obtain homozygous 3' UTR knockout cells, probably because MOLT-4 cells are of tetraploid karyotype so that it was difficult to completely delete all copies of 3' UTR-DNAs with CRISPR-Cas9 (Fig. 2a). Considering this,

our results might underestimate the effect of the 3' UTR on endogenous PD-1 expression (Fig. 2c-e)". (see p. 11 lines 7-11).

e: Please change the y-axis to logarithmic and calculate the half-life of PD-1 mRNAs with and without the full 3'-UTR. Please add error bars.

Reply: Done. See p. 10 lines 21-22 and Fig. 2e.

R1. mC -Figure 3

Please indicate the localization in the 3'-UTR using the nucleotide position within the full 3'-UTR not the P2 fragment throughout the study to avoid confusion with numbering in other figures (e.g., Figure 6).

Reply: Done.

d: The experiment/figure does not contribute to the manuscript. Either comment on the relevance of the figure or remove it from the manuscript.

Reply: Removed.

g and h: Exchange the position of g and h to match the data presented previously.

Reply: Done.

R1. mC -Figure 5

a: Please indicate the positions of P1, P2, and P3 in the scheme.

Reply: Done. See Fig.5 b.

b and c: How efficient is the knockout for the different RBPs targeted by CRISPR-Cas9 (Methods section: TIDE)?

Reply: Multiple gRNAs were designed for each targeting site, and only the ones with KO efficiency greater than 70% (evaluated with TIDE) were used for subsequent experiments. We added this information in the text (see p. 34 lines 17-18).

R1. mC -Figure 6

g: MBLN1 knockout increases proliferation in colon carcinoma cells. Is this related to PD-1 regulation by MBNL1? Please comment on the relevance to the main topic of the paper.

Reply: We added the comment that "Since the silencing of PD-1 could promote cancer cell proliferation, and we have demonstrated the knockout of MBNL1 reduced PD-1 expression, we were next wondering whether MBNL1 had effect on cancer cell proliferation by knocking out MBNL1 in WiDr cells. Indeed, depletion of MBNL1 significantly accelerated cell proliferation (Fig. 6h), suggesting that the MBNL1/PD-1 axis could potentially modulate cancer cell proliferation but more direct evidence is required." (see p. 22 lines 5-10).

R1. mC -Figure 7

b and e-h: The rhesus PD-1 3'-UTR contains a shorter stabilizing (CUG)_x portion as well as the destabilizing Rhe Elmt compared with the human PD-1 3'-UTR. Therefore, one would expect the rhesus UTR to be more destabilizing compared to the human UTR. This is not the

case (according to b). Please comment.

Reply: Because the (CUG)_x and the Rhe Elmt are not the only elements that affect mRNA stability, there could be other destabilizing elements that the human PD-1 3'-UTR contains but the rhesus does not. We added this comment in the main text (see p. 26 lines 10-15).

Reviewer #2 (Remarks to the Author):

In this study, Lai et al. use reporter assays and endogenous gene editing to elucidate the regulatory role of the 3' UTR of the gene encoding PD-1 (PDCD1) using sequences from human, mouse and several non-human primates. These studies reveal that while the full ~1kb 3' UTR sequence of PD-1 has a large regulatory effect that reduces expression of a reporter construct or the endogenous transcript, that this regulatory activity is distributed over a number of regions across the sequence, each of which has a relatively small effect in isolation. Additionally, they investigate the activity of a number of RNA binding protein genes that act in *trans* to regulate PD-1 expression level, and similarly discovered that no single factor has a dominant effect, with a number of factors having a subtle effect, suggesting a coordinate regulation of gene expression through the small contributions of a number of RNA binding proteins.

Overall, these findings are in line with our understanding of post-transcriptional regulation of gene expression, and form a valuable example of deep characterization of the contributions of individual regulatory elements in a single 3' untranslated region to work as an ensemble to regulate gene expression. Additionally, the fact that their experimental approach is informed by an evolutionary perspective adds value to their findings: the lower level of conservation of the 3' UTR relative to the coding sequence that the authors note in Figure 1 is a general characteristic of many 3' UTRs. Their observation of apparent functional activity despite an overall lower level of sequence conservation, and particularly their dissection of the contribution of variation in 3' UTR sequence in PDCD1 across mammals to regulatory activity (Figure 7) is appreciated. Finally, their choice of PD-1 as an example UTR to study in detail is apt. This gene is an important therapeutic target in its own right, and genetic variation altering its exact expression level may have important biological consequences, in line with the authors' hypothesis that the 3' UTR of PD-1 may be a locus undergoing rapid evolution that drives its sequence variation across species.

The study is well conducted and contains important controls that support the authors' main findings and interpretations and will be of value to immunologists and biologists interested in understanding post-transcriptional control of gene expression generally, or researchers seeking to conduct similar studies on their own gene of interest. As such, it merits publication in *Communications Biology*, but would be strengthened by a small number of focused experiments and some key revisions to the discussion:

R2.C1: In particular, the studies in Figures 2 and 3 suggest that while the endogenous 3' UTR has a moderately strong effect (~60% increase in PD-1 protein expression with the whole UTR KO in Fig. 2C), that large truncations had a strong regulatory effect in a reporter assay (Fig. 3C). Then, the data with small substitution mutations of individual regulatory elements had relatively small effect size. Therefore, it would be of interest to know what the effect of smaller individual deletions would be in regulating the endogenous expression of PDCD1 in a cell type that expresses this gene natively. This could be done, for example, using the 3' UTR dissection method that the senior author of this manuscript published <https://www.ncbi.nlm.nih.gov/pmc/articles/PMC5737544/>

Reply: Thank you for your constructive suggestion. We selected two representative small regions – the CUG repeat (283-362) and the E1 region (417-464) with enhancing and inhibiting activity respectively. After deletion with CRISPR-Cas9 in MOLT-4 cells, we measured the effect on endogenous PD-1 expression. As expected, deletion of CUG-repeat caused modest but significant decrease in PD-1 level; deletion of E1 region led to significant increase in PD-1 but the effect was smaller than deletion of the full PD-1 3' UTR. We added this new result to Fig. 6d and in the text (p. 21 lines 10-14).

R2.C2: Additionally, it would strengthen the paper to demonstrate that the regulation dissected throughout the manuscript has a biological impact on some measurable aspect of lymphocyte function (in donor T cells or Jurkat cells, for example) in line with the authors' hypothesis that 3' UTR sequence variation can alter immunological outcomes to an extent that it drives natural selection acting upon this locus.

Reply: To measure biological impacts, we examined the activation status of 3' UTR knock-out MOLT-4 cells (a T lymphoblast cell line) by measuring specific molecular markers. We found that the activity was inhibited in the knock-out cells compared to the cells with an intact 3' UTR. We also measured Ki67 expression and performed CCK 8 assays to detect the rate of cell proliferation. It was showed that the cells with loss of the PD-1 3' UTR proliferated much faster than control cells. Those distinct biological phenotypes suggested that the sequence variation of PD-1 3' UTR could potentially alter immunological outcomes. We added this new result to Fig. 2f-i and in the text (see p.10 line22 - p.11 line7).

R2.C3: Finally, in their discussion of the evolution of PD-1 the authors sometimes language that does not totally accurately capture the current understanding of PD-1 and its role in tumor biology. For example the claim that "pathological elevation of PD-1 is one of the major causes for cancer immune evasion" conflates T cell exhaustion with immune evasion. A careful examination of the claims made in this section with the current state of the field is warranted.

Reply: Done.

Reviewers' comments:

Reviewer #1 (Remarks to the Author):

The authors have satisfactorily addressed all my concerns. I have no further comments.

Reviewer #2 (Remarks to the Author):

I thank the authors for their follow up experiments.

The small deletion mutation experiments in the endogenous 3' UTR are concordant with their broader conclusions that individual regions of the PD-1 3' UTR have small effects that cumulatively regulate the turnover of this transcript.

The experiment describing the biological outcome of PD-1 3' UTR deletion raises several questions that I hope the authors can address. Specifically, the mechanism of action for increasing the proliferation rate of MOLT-4 cells in the steady state is unclear. What is the source of PD-1 ligands in this system? Does this cell line constitutively express PD-L1? Is the TCR signaling pathway constitutively active in this transformed cell line and are other molecules that PD-1 signaling would impact (e.g. SHP-2) constitutively expressed? Can the process of performing gene editing and selecting a sub-clone by itself lead to altered proliferation rates in this cell line?

Re: Revision of COMMSBIO-22-3584A

Point-by-point response to reviewers' comments:

Reviewer #1 (Remarks to the Author):

The authors have satisfactorily addressed all my concerns. I have no further comments.

Reply: Thank you for your satisfaction with the improvement of our work!

Reviewer #2 (Remarks to the Author):

I thank the authors for their follow up experiments.

The small deletion mutation experiments in the endogenous 3' UTR are concordant with their broader conclusions that individual regions of the PD-1 3' UTR have small effects that cumulatively regulate the turnover of this transcript.

The experiment describing the biological outcome of PD-1 3' UTR deletion raises several questions that I hope the authors can address. Specifically, the mechanism of action for increasing the proliferation rate of MOLT-4 cells in the steady state is unclear. What is the source of PD-1 ligands in this system? Does this cell line constitutively express PD-L1? Is the TCR signaling pathway constitutively active in this transformed cell line and are other molecules that PD-1 signaling would impact (e.g. SHP-2) constitutively expressed? Can the process of performing gene editing and selecting a sub-clone by itself lead to altered proliferation rates in this cell line?

Reply: Thank you for your recognition of our efforts at validating the concordance between the effect of the small deletions and the broader conclusion.

In terms of the altered proliferation rate of MOLT-4 cells, first we could probably be able to rule out the possibility that it was due to the selection of a sub-clone by itself, because we tested the other sub-clone that we did not show and observed a similar result.

To propose the mechanism of action for increasing the proliferation rate of MOLT-4 T cells, we checked the expression profile of MOLT-4 cells and found both PD-L1 and SHP-2 were expressed, so it is possible that the MOLT-4 cell line is constitutively activated. This, however, seems not to be the reason why deletion of PD-1 3' UTR promoted MOLT-4 cell proliferation, as it is well established that increased PD-1 level (due to 3' UTR deletion here) would inhibited, not promoted, T cell proliferation via this signaling pathway.

In fact, MOLT-4 cells are cancer cells derived from a patient with T-cell acute lymphoblastic leukemia. Recent studies showed that intrinsic PD-1 expression in many types of cancer cells such as melanoma (Sonja Kleffel et al., *Cell*, 2015), pancreatic cancer, hepatocellular carcinoma (Ning Pu et al, *Cancer letters*, 2019), lung cancer (Fanyi Gan et al., *Translational Cancer Research*, 2022), and skin cancer (Han Yao et al, *Frontiers in Immunology*, 2018) can promote cell proliferation. And one of the mechanisms is up-regulation of mTOR which modulates some of genes related to cell proliferation (Sonja Kleffel et al., *Cell*, 2015). We therefore hypothesized that deletion of PD-1 3' UTR could influence this pathway. We detected mTOR expression in PD-1 3' UTR-deleted MOLT-4 cells. Indeed, the mTOR expression was increased by 40%, suggesting PD-1 3' UTR deletion promoted MOLT-4 cell proliferation through the oncogenic pathway, not the immunity-regulating pathway. However, the detailed mechanism behind could be more complicated.

We added this new result to Fig. 2j and in the text that *“We were next interested in the mechanism by which deletion of PD-1 3' UTR increased the proliferation rate of MOLT-4 cells. MOLT-4 cells are cancer cells derived from a patient with T-cell acute lymphoblastic leukemia. Recent studies showed that intrinsic PD-1 expression in many types of cancer cells can promote cell proliferation, including melanoma³⁵, pancreatic cancer, hepatocellular carcinoma³⁶, lung cancer³⁷, and skin cancer³⁸, and one of the mechanisms is up-regulation of mTOR which modulates some of genes related to cell proliferation³⁵. We therefore detected mTOR expression in PD-1 3' UTR-deleted MOLT-4 cells. Indeed, the mTOR expression was increased by 40%, suggesting PD-1 3' UTR deletion promoted MOLT-4 cell proliferation through the oncogenic pathway. However, the detailed mechanism could be more complicated and further investigation would be valuable.”* (see page 10 lines 2-3; page 11 lines 6-15). Thank you!